# Comparative Quality and Trend of Remotely Sensed Phenology and Productivity Metrics across the Western United States

**Ethan E. Berman** [1,*], **Tabitha A. Graves** [1], **Nate L. Mikle** [1], **Jerod A. Merkle** [2], **Aaron N. Johnston** [3] and **Geneva W. Chong** [3]

1   U.S. Geological Survey, Northern Rocky Mountain Science Center, Glacier Field Station, 38 Mather Drive, West Glacier, MT 59936, USA; tgraves@usgs.gov (T.A.G.); natemikle@gmail.com (N.L.M.)
2   Department of Zoology and Physiology, University of Wyoming, Department 3166, 1000 E University Ave, Laramie, WY 82071, USA; jmerkle@uwyo.edu
3   U.S. Geological Survey, Northern Rocky Mountain Science Center, 2327 University Way, Suite 2, Bozeman, MT 59715, USA; ajohnston@usgs.gov (A.N.J.); Geneva_chong@usgs.gov (G.W.C.)
*   Correspondence: bermane@gmail.com

**Abstract:** Vegetation phenology and productivity play a crucial role in surface energy balance, plant and animal distribution, and animal movement and habitat use and can be measured with remote sensing metrics including start of season (SOS), peak instantaneous rate of green-up date (PIRGd), peak of season (POS), end of season (EOS), and integrated vegetation indices. However, for most metrics, we do not yet understand the agreement of remotely sensed data products with near-surface observations. We also need summaries of changes over time, spatial distribution, variability, and consistency in remote sensing dataset metrics for vegetation timing and quality. We compare metrics from 10 leading remote sensing datasets against a network of PhenoCam near-surface cameras throughout the western United States from 2002 to 2014. Most phenology metrics representing a date (SOS, PIRGd, POS, and EOS), rather than a duration (length of spring, length of growing season), better agreed with near-surface metrics but results varied by dataset, metric, and land cover, with absolute value of mean bias ranging from 0.38 (PIRGd) to 37.92 days (EOS). Datasets had higher agreement with PhenoCam metrics in shrublands, grasslands, and deciduous forests than in evergreen forests. Phenology metrics had higher agreement than productivity metrics, aside from a few datasets in deciduous forests. Using two datasets covering the period 1982–2016 that best agreed with PhenoCam metrics, we analyzed changes over time to growing seasons. Both datasets exhibited substantial spatial heterogeneity in the direction of phenology trends. Variability of metrics increased over time in some areas, particularly in the Southwest. Approximately 60% of pixels had consistent trend direction between datasets for SOS, POS, and EOS, with the direction varying by location. In all ecoregions except Mediterranean California, EOS has become later. This study comprehensively compares remote sensing datasets across multiple growing season metrics and discusses considerations for applied users to inform their data choices.

**Keywords:** phenology; productivity; forage; green up; growing season; phenocam; timing; land surface phenology; elk; deer

## 1. Introduction

Vegetation phenology and productivity are important drivers of ecosystem function by influencing processes as varied as surface energy balance [1] and plant species distribution [2]. Shifts in the timing of plant seasonality occur largely due to annual weather patterns and climate change [3],

and these dynamics have consequences throughout terrestrial ecosystems. For example, phenology and productivity patterns strongly influence animal behavior, survival, and population dynamics [4]. Shifts in the timing and amount of vegetation can lead to trophic mismatch [5] and put stress on migratory species resilience and adaptive capacity [6]. Thus, to understand changes to the scale, rate, spatial configuration, and variability of ecological processes, we need to accurately measure vegetation phenology and productivity at broad spatial and temporal scales [7].

In recent decades, land surface phenology (LSP), the study of vegetation phenology and productivity from remote sensing, has revolutionized our understanding of ecological responses to phenological change [8]. LSP observations broaden the spatial scale of data to previously unattainable extents, enabling analyses of regional and continental vegetation patterns over time, with certain datasets extending back more than 30 years. Studies of LSP metrics in recent decades in North America have shown trends toward later senescence in the fall but inconclusive evidence for earlier spring green-up [9]. Green-up trends within ecological communities are complex, with plant species in the same community often showing opposite responses in both sign and magnitude to general warming patterns [10]. These trend analyses, along with annual LSP metrics, provide users (researchers, biologists, ecologists, natural resource managers, etc.) from a variety of fields with important insights into ecosystem processes and changing landscape dynamics driven by weather, climate, and human and natural disturbances. However, few studies have provided regional summaries of change that are accessible to users who are managing resources [7,11] or provided information on trends in variability of phenology over time. Insights into changing variability is important for wildlife researchers, as environmental predictability may shift habitat use, population density, and movement patterns [12].

Wildlife biologists began using LSP metrics in the early 2000s to better understand the influence of vegetation on the dynamics and distribution of animal populations including birds [13,14] and ungulates such as elk and deer [4,8,15–19]. For example, the timing of spring has implications for the fitness and body condition of ungulates [16,20–22] and peak instantaneous rate of green-up date (PIRGd), the date half way between start of season (SOS) and peak of season (POS) and representative of optimal forage quality, explains migratory patterns of ungulates surfing the green wave [23–25]. In addition, spatial heterogeneity of plant phenology, which may be declining due to warming temperatures, relates to the reproduction rates of caribou [26]. Autumn phenology, which has received considerably less attention than that of spring, can influence body mass and overwinter survival of mule deer [4] and migratory patterns of elk and red deer [18,27,28]. In addition to phenology, vegetation productivity is closely correlated to greenness indices such as Normalized Difference Vegetation Index (NDVI) [8] and explains ungulate habitat use [24,29], health characteristics [30], and demographic parameters [16].

Changes in the seasonal timing of LSP metrics and the development of new datasets from various satellite platforms have received substantial focus from the remote sensing field [9,31–34]. However, despite the importance of LSP metrics to ecological applications, few studies have tested the quality of competing datasets in measuring LSP metrics against ground or near-surface observations or examined their relative agreement across various land cover types (but see [32,35]). Because LSP metrics synthesize information from millions of individual plants and multiple species, large biases across diverse vegetation communities can occur, and metrics may not directly represent the biological processes of the vegetation of interest [36]. Differences in the processing algorithm of LSP data, such as the use of logistic curve fitting techniques versus splines, can also greatly impact performance [37]. Although users choose datasets based on their desired temporal and spatial scale, often several datasets exist with similar resolution yet different processing methodologies. A comparison of the quality of freely accessed and commonly derived LSP datasets against near-surface observations would assist users in selecting the best dataset for their application.

Here, we provide a comparison of commonly used phenology and productivity metrics derived from 10 freely available LSP datasets. We examine correlation and bias in the western United States from 2002 to 2014 with near-surface observations from PhenoCam, a network of cameras spread

throughout North America providing data multiple times per day [38]. We provide users of phenology and productivity metrics derived from remote sensing an indication of the strengths and weaknesses of different datasets, especially as related to land cover. We also compare long-term (1982–2016) trends in phenology from two leading LSP data products to identify spatial patterns, assess the agreement about changing vegetation dynamics over time, and describe changing variability in phenology.

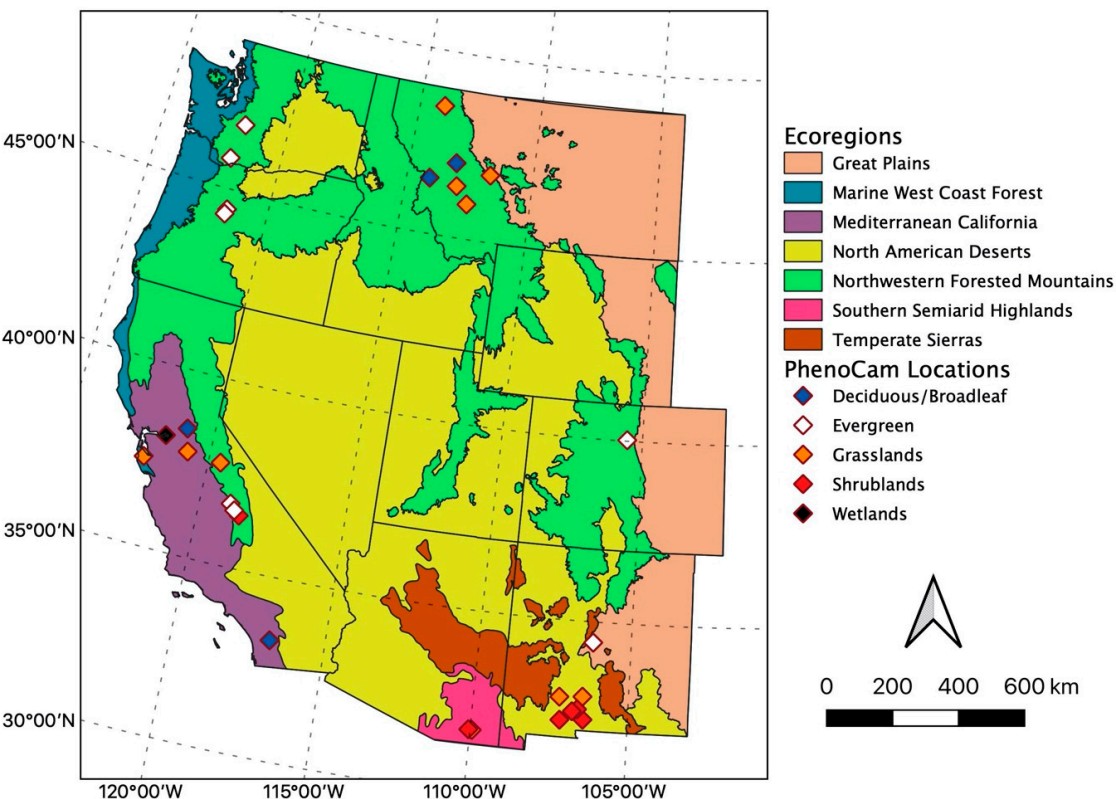

**Figure 1.** The western United States (11 states) overlaid with Level 1 Ecoregions [39] and the locations and vegetation types of 29 PhenoCam sites. Note that some PhenoCam locations are slightly jittered to prevent overlap and therefore may not represent exact location (but are always in same ecoregion).

## 2. Materials and Methods

We evaluated six phenology and three productivity metrics across the western United States, which represents a diverse range of ecosystems and land cover types (Figure 1). To evaluate agreement with PhenoCam data, we focused on 10 datasets from 2002 to 2014 (Table 1), comparing overall agreement and agreement within land cover types. We then assessed long-term trends using a subset of more strongly associated datasets from 1982 to 2016.

Phenology and productivity metrics can be derived from reflectance values recorded in optical satellite imagery. Optical satellite imagery captures differences in the reflectance of vegetation through phenology cycles. Most commonly users measure vegetation indices as a ratio between reflectance in the visual and near-infrared parts of the electromagnetic spectrum, with slightly different formulas for NDVI and Enhanced Vegetation Index (EVI; [40]). From curves describing the cycle of change, users extract and evaluate key LSP phenology and productivity metrics including those evaluated here (Figure 2): SOS, PIRGd, POS, end of season (EOS), length of spring (LOSp), length of growing season (LGS), integrated vegetation index (IVI), peak vegetation index (PVI), and amplitude of vegetation index (AVI).

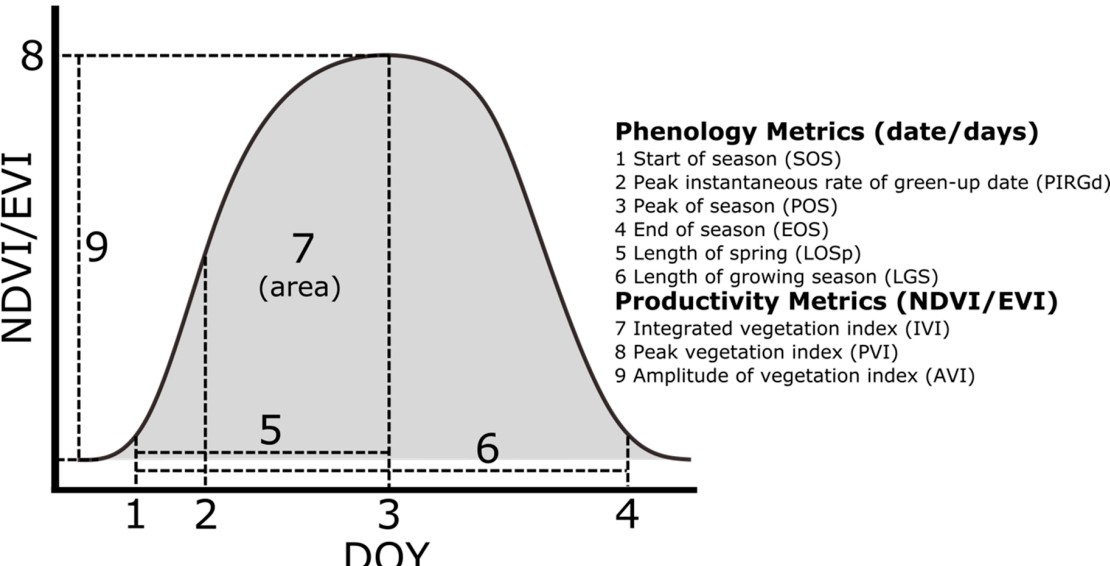

**Figure 2.** Six phenology and three productivity metrics shown through an example growing season by day of year (DOY). Note that these metrics are described through values of Normalized Difference Vegetation Index (NDVI)/Enhanced Vegetation Index (EVI) but certain datasets may produce metrics using other methods/algorithms.

Most datasets we evaluated use NDVI or EVI calculated from the surface reflectance of the Moderate Resolution Imaging Spectroradiometer (MODIS) or Advanced Very High Resolution Radiometer (AVHRR) sensors (Table 1). We also evaluated one dataset, the National Phenology Network first leaf spring index, hereinafter referred to as NPN, that models spatially explicit temperature measurements parametrized via an extensive network of in situ phenological observations [41]. This is the only dataset evaluated that does not incorporate optical satellite imagery, but we included it because it is readily available and provides annual SOS estimates across the contiguous United States (CONUS). We also assessed two net primary productivity (NPP) datasets (CONUS Landsat NPP and CONUS MODIS NPP) that combine remote sensing with meteorological data and plant physiological parameters [42]. Because they represent the total productivity across the season, we include NPP in the IVI comparisons. AVI and PVI both measure the height of the curve at the peak, but AVI only includes the height from the baseline to the peak. Spatial resolutions spanned 30 m to approximately 5 km (0.05 degrees) and all data were freely available. Most datasets required minimal code and memory because the relevant phenology and productivity metrics were pre-processed and could be downloaded directly (e.g., SOS, POS, and EOS; Table 1). When necessary, we derived metrics (e.g., PIRGd and LOSp) from available metrics (see Appendix A Table A1 and Equations (A1)–(A7) for a complete description of metric availability, source, and derivation formula). We calculated all metrics for one dataset based on MODIS MOD09Q1 NDVI (referred to as DLC for "double logistic curve"), fitting a double logistic curve and accounting for snow cover following methods of [24], because of its use in many wildlife applications.

**Table 1.** Abridged data table summarizing the 10 freely available datasets evaluated against PhenoCam. See Table A1 and Equations (A1)–(A7) for more information including methods used to derive individual metrics and data acquisition information.

| Dataset | Base Input | Spatial Resolution | Temporal Coverage | Accessible Online? | Significant Post-processing Time Requirement? | Under Production? | Spatial Coverage | Reference |
|---|---|---|---|---|---|---|---|---|
| AVHRRP | NDVI | 1 km | 1989–2014 | yes | no | no | CONUS | [43] |
| eMODIS | NDVI | 250 m | 2001–2018 [1] | yes | no | yes | CONUS | [44] |
| MCD12Q2 | EVI | 500 m | 2001–2017 | yes | no | yes | Global | [45] |
| CMGLSP | EVI | 0.05° | 1982–2016 | no | no | yes | Global | [46] |
| VIPPHEN-EVI2 | EVI | 0.05° | 1981–2016 | yes | no | no | Global | [47] |
| VIPPHEN-NDVI | NDVI | 0.05° | 1981–2016 | yes | no | no | Global | [47] |
| NPN (First Leaf Spring Index) | Temp, in situ observations | 0.0417° | 1981–2019 | yes | no | yes | CONUS | [41] |
| Landsat NPP | NDVI, meteorological inputs | 30 m | 1986–2018 | yes | no | yes | CONUS | [42] |
| MODIS NPP | NDVI, meteorological inputs | 250 m | 2001–2018 | yes | no | yes | CONUS | [42] |
| DLC | NDVI | 250 m | 2000–present | yes | yes | yes | Global | [23] |
| PhenoCam | Green chromatic coordinate | Point | 2000–present | yes | no | yes | U.S. and Canada [2] | [48] |

[1] Collection 6 of eMODIS starts in 2003, but Collection 5 data is available from 2001. This analysis used Version 5 for 2002; [2] Phenocam also has a limited number of sites in Panama, Hawaii, and Europe

To compare datasets, we evaluated the agreement of LSP metrics from each dataset with metrics from PhenoCam data [48] at 29 sites (106 site-years) across the western United States (Figure 1) from 2002 to 2014. Using the R package phenocamr [49], we downloaded 3 day window PhenoCam data and calculated the 90th percentile green chromatic coordinate (GCC; Equation (1)). The 90th percentile value effectively accounts for day-to-day variation due to weather and illumination patterns [50]. We defined rising and falling phenophases with a 10% amplitude threshold (to derive SOS and EOS) as described in PhenoCam documentation [48,50]. We chose the 10% threshold because it more closely resembles SOS than a 25% or 50% threshold and minimizes the bias between PhenoCam transition dates and MODIS transition dates [35]. GCC is a vegetation index derived from RGB camera imagery and is defined as:

$$GCC = \frac{DNG}{(DNR + DNG + DNB)} \tag{1}$$

where DNG, DNR, and DNB represent the digital number (i.e., strength) of the green, red, and blue channels, respectively.

To assess the agreement of LSP metrics with PhenoCam, we compared correlation using the coefficient of determination ($R^2$) and the magnitude of difference using mean bias. These common measures of statistical agreement have been previously used to compare LSP and near-surface phenology [36,51,52]. Mean bias is defined as:

$$\text{Mean bias} = \frac{1}{N} \sum_{i=1}^{N} (\text{est}_i - \text{obs}_i) \tag{2}$$

where $\text{est}_i$ and $\text{obs}_i$ are the ith estimate (from LSP dataset) and observation (from PhenoCam) respectively. For productivity metrics, scales differ across datasets and thus we focused on correlation. We compared both overall agreement and agreement by landcover type, using the following vegetation types defined for PhenoCam sites (PhenoCam metadata: grasslands (GR, 45 site-years), shrublands (SH, 7 site-years), deciduous/broadleaf (DB, 27 site-years), evergreen (EN, 25 site-years), and wetlands (2 site-years). We excluded two sites in the Mediterranean California ecoregion that displayed a non-northern-temperate seasonal signal (SOS > 225) because green-up begins in late fall with rainfall and ends in early spring with drought. We excluded wetlands from the analysis by land cover classification due to the limited number of sites.

We analyzed long-term trends (1982–2016) across the western U.S. using CMGLSP and VIPPHEN_EVI2, which agreed best with PhenoCam of the datasets that extended back to the 1980s. Of the six phenology metrics, we reported trends for SOS, POS, and EOS, because the other three are derived directly from these metrics and PIRGd spatial patterns are similar to SOS and POS. Using the Theil–Sen slope, the median of slopes between all pairs of observations within a pixel [53], we assessed each metric by pixel, including only pixels with data for at least 18 of 35 years. As a non-parametric test the Theil–Sen is more robust to outliers and provides higher statistical power when non-normality exists [54]. Negative slopes indicate change to earlier dates and positive slopes indicate change to later dates. We did not screen pixels for disturbances as the goal of this work is to identify patterns and magnitudes of change, rather than make inferences about the causes of phenological change.

We evaluated overall variation within each pixel based on the standard deviation for each metric and whether variability has increased across years by applying the Theil–Sen slope to the absolute values of the residuals of the trend estimate against time. In addition, we measured consistency between the two datasets, defined as the by-pixel agreement that phenology dates are trending earlier or later. Lastly, to assess agreement in regional patterns of change, we report the proportion of area where both datasets agreed on direction of change by ecoregions using the United States Environmental Protection Agency Level 1 Ecoregions of North America dataset [39]. We used the statistical computing environment R for all analyses [55] and R package rkt [56] to calculate the Theil–Sen slope estimator.

## 3. Results

### 3.1. Agreement of LSP metrics with PhenoCam

We compared LSP metrics to 106 site-years of PhenoCam data from 29 sites. We found substantial variation in agreement between remotely sensed datasets and near-surface observations (Figure 3; Figure 4; Appendix B Tables A2–A6 for full results). Across datasets and six phenology metrics, $R^2$ ranged from 0.03 to 0.37 (SOS), 0.20 to 0.55 (PIRGd), 0.25 to 0.54 (POS), 0.16 to 0.45 (EOS), 0.01 to 0.11 (LOSp), and 0 to 0.10 (LGS). Absolute value of mean bias (in days) ranged from 4.39 to 25.49 (SOS), 0.38 to 17.66 (PIRGd), 7.05 to 23.82 (POS), 0.52 to 37.92 (EOS), 12.45 to 44.66 (LOSp), 3.78 to 58.47 (LGS). Mean bias was lowest for SOS and PIRGd. Four datasets matched SOS and PIRGd from PhenoCam within 7 days (MCD12Q2, VIPPHEN_EVI2, DLC, and eMODIS for SOS; CMGLSP, MCD12Q2, VIPPHEN_EVI2, and VIPPHEN_NDVI for PIRGd). Generally, $R^2$ values were higher for PIRGd and POS than SOS and EOS. Datasets were least correlated with PhenoCam transition dates for LOSp and LGS. In pairing a high $R^2$ value with a low mean bias, MCD12Q2 and CMGLSP had best overall agreement with PhenoCam.

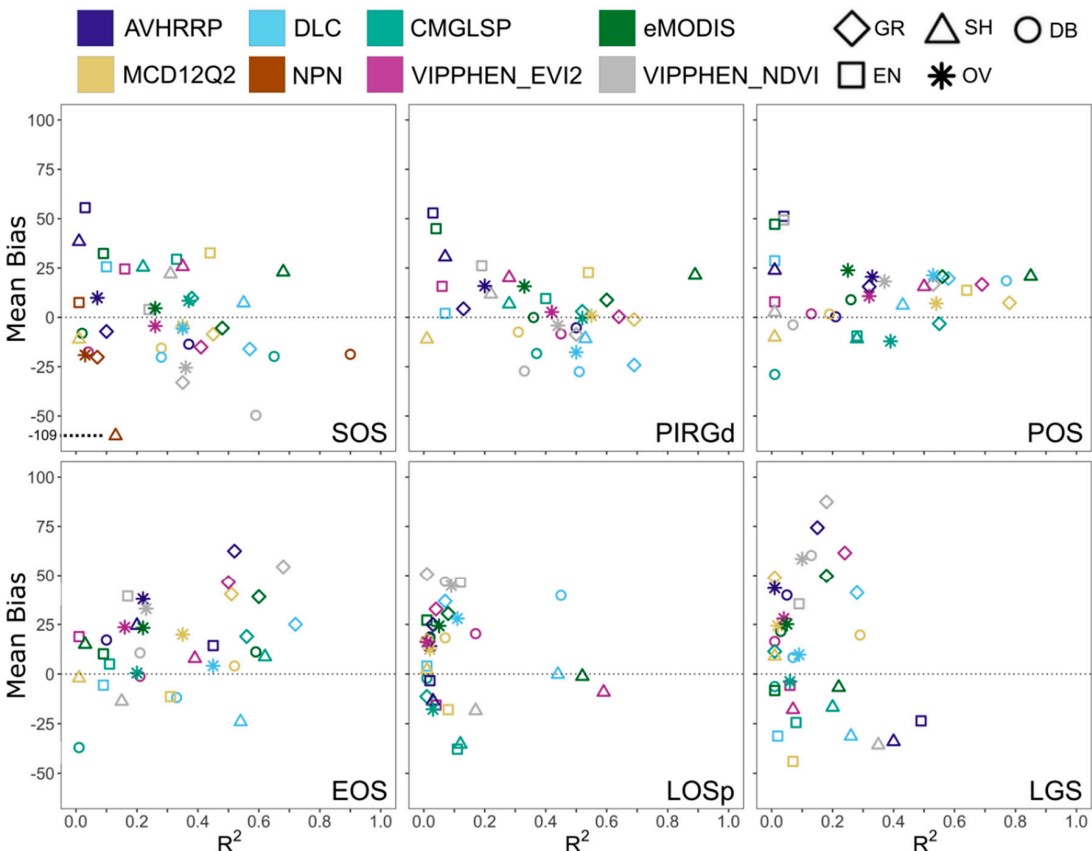

**Figure 3.** $R^2$ and mean bias (in days) for six phenology metrics classified by land cover type identified in PhenoCam metadata and overall for all data points. The phenology metrics are start of season (SOS), peak instantaneous rate of green-up date (PIRGd), peak of season (POS), end of season (EOS), length of spring (LOSp) and length of growing season (LGS). The classifications are grasslands (GR), shrublands (SH), deciduous/broadleaf (DB), evergreen (EN) and overall (OV). In general, this illustrates how the overall evaluation falls between the landcover-specific values for each dataset and metric combination.

Phenology metrics had large differences in correlation and bias by land cover type (Figure 3). Most datasets agreed moderately well with PhenoCam in grasslands, shrublands and deciduous/broadleaf forests for the four key phenology dates (SOS, PIRGd, POS, and EOS). Duration

metrics (LOSp and LGS) correlated poorly with PhenoCam across all datasets. At grassland sites ($n = 45$), dates for most datasets correlated well with PhenoCam dates. At shrubland sites ($n = 7$), eMODIS was best for SOS, PIRGd, and POS, whereas CMGLSP was best for EOS. At deciduous/broadleaf sites ($n = 27$), CMGLSP and NPN agreed best for SOS, DLC for POS, and MCD12Q2 and eMODIS for EOS. Most datasets agreed well for PIRGd in deciduous/broadleaf forests. Datasets generally showed poor agreement with PhenoCam at evergreen forest sites ($n = 25$), with the exception of MCD12Q2.

Overall, datasets better represented phenology than productivity metrics (Figure 4). Across datasets, $R^2$ for the three productivity metrics ranged from 0.00 to 0.16 (IVI), 0.00 to 0.15 (PVI) and 0.00 to 0.34 (AVI), with low values in overall $R^2$ masking high correlations in a few land cover types. For the IVI metric, MODIS_NPP had an $R^2$ value of 0.9 at deciduous/broadleaf locations and VIPPHEN_EVI2 and VIPPHEN_NDVI both agreed well. MCD12Q2 was highly correlated with PhenoCam at evergreen locations for PVI and AVI.

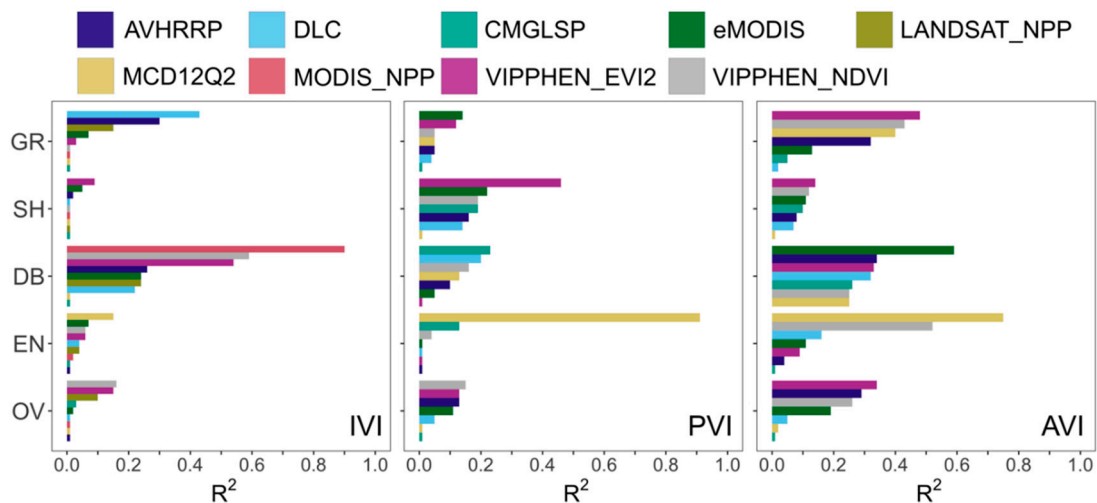

**Figure 4.** $R^2$ for three productivity metrics classified by land cover type identified in PhenoCam data and overall for all data points. The productivity metrics are integrated vegetation index (IVI), peak vegetation index (PVI) and amplitude of vegetation index (AVI). The classifications are grasslands (GR), shrublands (SH), deciduous/broadleaf (DB), evergreen (EN) and overall (OV).

### 3.2. Historical Trend Analysis for Western United States

Of the 10 datasets evaluated, five extend back to 1982 (CMGLSP, VIPPHEN_EVI2, VIPPHEN_NDVI, AVHRRP and NPP) and therefore provide an appropriate historical record for relatively long-term trend analysis. Based on the comparison with PhenoCam, we chose to analyze CMGLSP and VIPPHEN_EVI2, plus NPN for SOS (NPN results in Appendix C Figures A2–A4). CMGLSP most often had the ideal combination of high $R^2$ and low mean bias across land cover types and VIPPHEN_EVI2 also agreed well with PhenoCam, especially in grasslands and shrublands.

Long-term trends indicate large changes in key phenology dates in some places by more than 70 days (2 days per year for 35 years) between 1982 and 2016 (Figure 5; see Figures A1 and A5–A7 for mean and standard deviation maps and histograms of changes). Trend patterns had high spatial heterogeneity, with substantial interspersion of pixels with delayed and advanced phenology in both datasets and all metrics. The strength of the trends varied spatially, but in both datasets the areas of greatest change in date occurred in New Mexico and Arizona (later SOS), low-lying areas of California (earlier EOS), and the Great Basin (earlier SOS, mixed strong trends for EOS). EOS in the Southwest was also generally later but was more variable in California. Higher elevation areas across Montana, Washington, Idaho and Colorado showed trends toward later SOS but mixed results for EOS. Over time, variation in phenology dates slightly increased across most regions, with very large increases in variability corresponding to areas with larger changes in date, namely the Southwest for



SOS and POS, and scattered areas around the edges of the Great Basin as well as California for EOS. The spatial patterns of agreement on the direction of trend across CMGLSP and VIPPHEN-EVI2 were also relatively heterogenous, with the least consistency occurring in Wyoming for SOS and in the Great Basin for EOS.

The consistency results over the entire study region showed ~60% agreement in trend direction for all three metrics (Figure 6). For SOS and POS ~30% of pixels agreed on earlier and later dates in trends, whereas for EOS, 44.4% of pixels in both datasets agreed on a later EOS (16.2% agreed on an earlier EOS). In assessing consistency by ecoregion, we found agreement between the datasets in the range of 50–70% of pixels for most of the regions and metrics as well as agreement on the predominant direction of change within ecoregions (Figure 6). For SOS, the two datasets agreed that more area trended towards earlier dates in Marine West Coast Forests, Mediterranean California, North American Deserts, and Northwestern Forested Mountains, and later dates in Great Plains, Southern Semiarid Highlands, and Temperate Sierras. For POS, the datasets agreed on more area trending toward earlier dates in Marine West Coast Forests, Mediterranean California, and North American Deserts, and later dates in Great Plains, Northwestern Forested Mountains, Southern Semiarid Highlands, and Temperate Sierras. We found the most dataset agreement across ecoregions for EOS, in which all regions agreed on more area trending towards later dates except Mediterranean California.

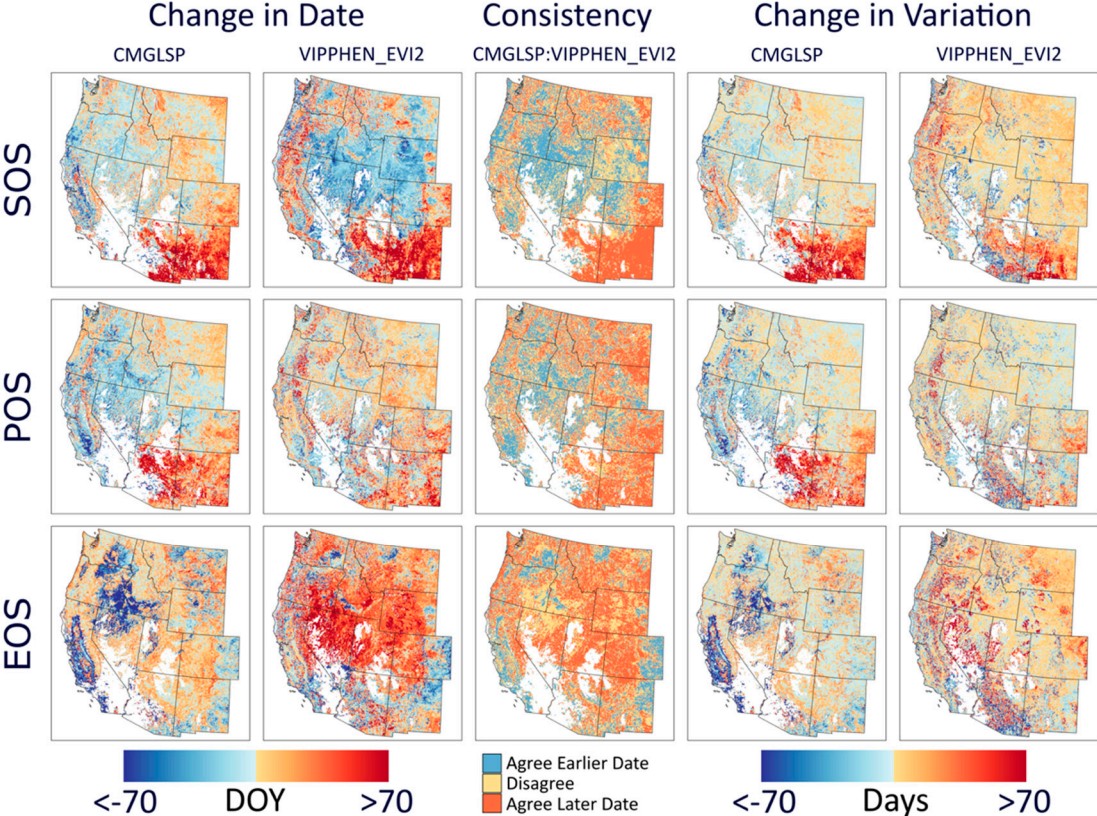

**Figure 5.** Change in date, consistency, and change in variation of CMGLSP and VIPPHEN_EVI2 across three phenology metrics from 1982 to 2016. Negative values correspond to earlier dates (or fewer days) and positive values to later dates (or more days). Consistency shows agreement in the direction of change between datasets. Change in date is the regressed trend slope multiplied by the number of years. Change in variation is the regressed trend slope, multiplied by the number of years, of the absolute value of the residuals of the trend estimate against time.

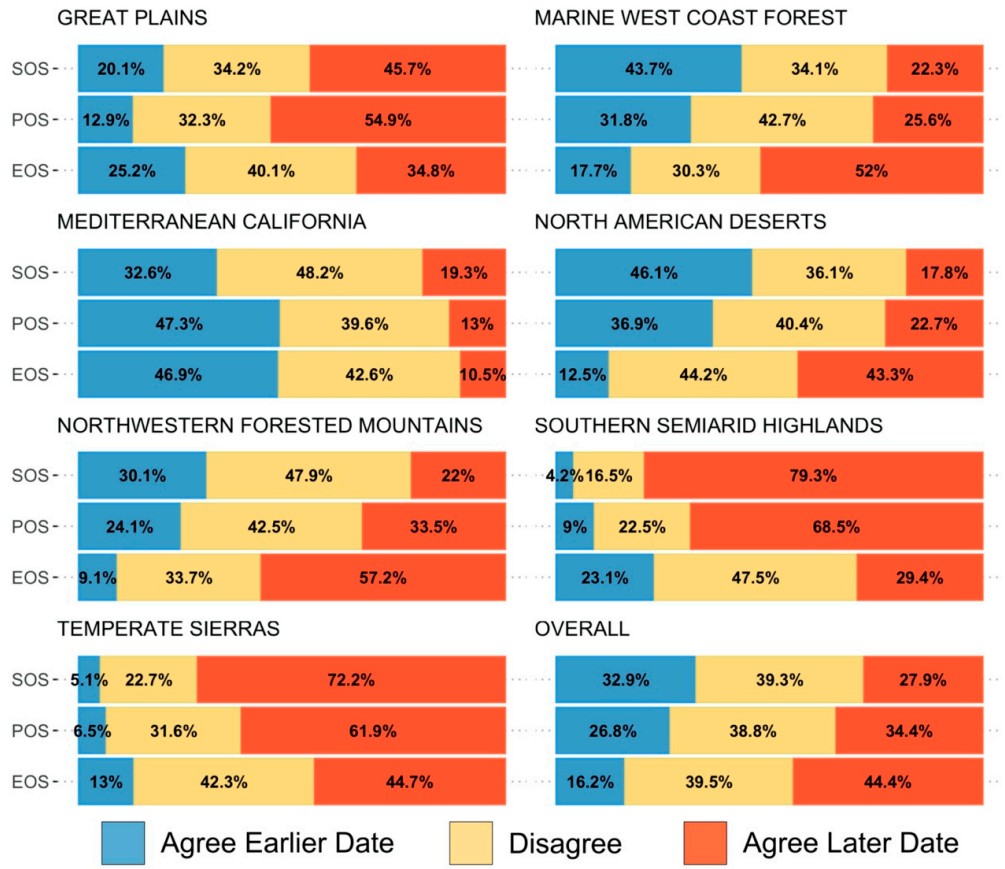

**Figure 6.** Consistency of CMGLSP and VIPPHEN_EVI2 long-term (1982–2016) trend direction over the entire study area and classified by ecoregion.

## 4. Discussion

In our comparison of 10 leading LSP datasets against a network of PhenoCam near-surface cameras, we found promising results in terms of $R^2$ and mean bias in some landcover types, but substantial variation between datasets and metrics (similar to [32] for SOS estimates). This highlights a need for improved communication and more extensive dialogue on the strengths, weaknesses and development goals of LSP datasets, especially with the proliferation of applied data users from a variety of research and management practices.

When choosing an appropriate dataset for analysis, applied users should consider study location, years needed, spatial resolution, processing capacity, and quality in different land cover classes. For phenology metrics, our results indicate MCD12Q2 and CMGLSP best matched PhenoCam observations derived using a 10% amplitude threshold and the 90th GCC percentile (Figure 3). For those interested in phenology prior to the 2000s, CMGLSP has global coverage and extends back to 1982, but has a coarse 0.05 degree (~5 km) spatial resolution and can only be acquired by request. VIPPHEN_EVI2 has the same temporal coverage and spatial resolution, can be downloaded directly, and agreed almost as well. For users working with high resolution data (such as from GPS collars in the wildlife field) and conducting analyses after 2001, MODIS-based datasets at 250 to 500 m spatial resolution provide finer-scale observations. MCD12Q2, with 500 m spatial resolution and global coverage, agreed best with PhenoCam and was the only dataset that had high correlations in evergreen forests. eMODIS also agreed well and has 250 m spatial resolution but is only available in the CONUS. The DLC method can be applied globally (we used MOD09Q1 as the base input) but requires substantial processing which may be prohibitive depending on study area size. This method also provides daily values of NDVI and instantaneous rate of green-up (IRG), which are useful for certain movement ecology and habitat questions such as those relating to the green-wave hypothesis [21,23–25,57].

Readily available datasets from MODIS (e.g., MOD13A1, MOD13Q1, and eMODIS NDVI) only provide cleaned and pre-processed weekly to bi-weekly NDVI/EVI values and therefore could provide a coarser resolution IRG time series.

Based on the differences in dataset correlation and bias we found between land cover types, users should consider the predominant land cover of their study area in choosing a phenology or productivity dataset (Figure 3; Figure 4). For instance, eMODIS agreed best for PIRGd in shrublands, yet MCD12Q2 was better in grasslands and evergreen forests. Large differences in correlation were observed between productivity datasets, which correlated poorly overall but well in certain landcover types. For IVI, DLC agreed well in grasslands, but MODIS_NPP agreed best in deciduous/broadleaf forests. The lowest correlations between LSP estimates and PhenoCam observations were in evergreen forests, where identifying phenological metrics is challenging because of the small annual change in greenness amplitude and over-saturation [35,58]. Using the red chromatic coordinate to process PhenoCam transition dates, as opposed to GCC, has shown potential to improve predictions of SOS and EOS in these environments [59]. Of all the LSP datasets, MCD12Q2 agreed best with PhenoCam at evergreen sites, possibly because it is EVI based [40]. Ecologists studying wildlife in areas with multiple land cover types should be aware that phenology metrics in some land cover types are more reliable than others, which could influence results for analyses comparing wildlife GPS locations across different land cover classes [17,60,61]. Quality of LSP metrics may differ geographically, and therefore our results are not necessarily indicative of dataset quality in other parts of North America or around the world [62].

Quantifying phenology and productivity trends through time is crucial to improve our understanding of ecosystem processes, such as how changing forage in critical habitat or management units affects wildlife and how to manage for future climate and phenology scenarios [8,26,63,64]. We found a 44.4% agreement between CMGLSP and VIPPHEN_EVI2 trends toward a later EOS across the Western U.S and only a 16.2% agreement toward an earlier EOS, which is consistent with other studies signaling a general pattern toward a later EOS [9,65,66]. Trends are influenced by a variety of factors, including temperature and precipitation, species succession, human and natural disturbance, and land management practices (as observed by [37] from 1982 to 2006). The complex interactions between these factors make it difficult to interpret the ecological significance of large and spatially heterogeneous trends, especially across diverse ecosystems.

The large phenology trends and high variability we observed over a 35 year period likely reflect real changes in remote sensing metrics, but highlight the need to understand what exactly is changing throughout time, the role of data processing choices, and whether the changes are biologically significant. We know that patterns of temperature and precipitation in some areas are complex, leading to variability in the timing, seasonality, and spatial heterogeneity of snow coverage [67], soil moisture, drought, and storms that may all influence the timing, variability, and heterogeneity of vegetation phenology and productivity. Variability can be an important component to ecosystem processes and to understanding the degree and mechanism of trends. For instance, vegetation strongly affected by climate conditions, such as invasive grasses responding to rainfall, can cause high year-to-year variability and therefore introduce uncertainty into the direction of overall trend, as well as have large effects on the ecosystem [68,69]. In our results, some regions in which LSP metrics changed by more than 70 days often showed high and increasing variability. Though these trends may be showing real changes in phenology throughout time within a pixel, it is challenging to interpret whether these changes are driven by variability caused by weather, invasive species, or other factors, or actually represent the changing dynamics of vegetation of interest, such as important forage species. This overall assessment is a first step in understanding the overall patterns and degree of change across the CONUS.

Several challenges exist in producing accurate LSP estimates and assessing their quality. Large differences in LSP metrics and trend are related to sensor-specific characteristics, such as revisit time and spectral resolution, as well as processing algorithms used to extract metrics, which may include

cloud and atmospheric correction, gap filling and curve fitting. Greenness (NDVI/EVI/GCC) values differ between satellite platforms, as well as with near-surface cameras, and are therefore not always comparable without applying translation equations [70]. Future work may scale PhenoCam and satellite derived productivity metrics to enable a comparison of mean bias and other statistics. Differences in consistency (by-pixel agreement on trend direction) between CMGLSP and VIPPHEN_EVI2 result solely from different processing algorithms, as both CMGLSP and VIPPHEN_EVI2 are based on EVI2 calculated from AVHRR and MODIS at a 0.05 degree spatial resolution (for CMGLSP, see [46]; for VIPPHEN_EVI2, see [47]). These two datasets yielded conflicting yet large trends in phenology around the Great Basin and Eastern Washington and Oregon, specifically for EOS. These areas experienced large fires, landcover changes, and other disturbances. It is possible that the different approaches to fitting NDVI/EVI curves and extracting metrics for these datasets are affected differently by these landcover type changes and thus lead to these conflicting signals.

In general, LSP datasets use different processing methodologies and different threshold values [36,37,71], and dataset development goals rarely correspond to specific biological events important to data users. For instance, biological events recorded by a botanist to mark the start of growing season may include first leaf or flower dates, whereas an LSP dataset such as MCD12Q2 captures SOS as the date when EVI crosses the 15% threshold of AVI [45]. Ref. [32] used in situ first leaf observations to validate SOS from various LSP datasets and found high root mean squared error (~20 days), indicating a need to better understand the link between biological events and image reflectance values. Our use of near-surface cameras removes variability that may occur when comparing LSP metrics with the timing of biological processes from in situ observations. Although some research assesses LSP quality through comparison with other kinds of ground-based datasets [72,73], this may confound differences resulting from discrepancies between biological events and their reflectance values with differences in image quality, which can vary inter-annually and regionally [62].

Datasets also vary in how they deal with growing seasons that span calendar years or have multiple annual cycles. Most datasets calculate phenology metrics based on a continuous time series yet report metrics in discrete annual data layers. The one exception in our analysis is the DLC dataset, which fits a curve to a single year of data. When phenology cycles span calendar years, such as when SOS occurs in the fall and/or when EOS occurs in late winter/early spring, users should ensure they understand the approach used to store the metrics within an annual data layer. Datasets handle patterns of multiple or missing annual wetness cycles in different ways, with some datasets reporting up to three annual cycles while others only report phenology dates from a single cycle that is not indexed to a general time of year. We most commonly see multiple annual cycles of green-up in arid and monsoonal environments, such as in the Southwest [74] and Mediterranean California, but this varies by year.. When the number of cycles varies, such as when drought years lead to missed wet periods [75], or seasonal weather patterns dictate two to three annual cycles, analyses of trends by year, as we have conducted, could suggest large changes in dates that might not reflect the complexity of the ecological impacts well. We briefly evaluated these potential issues, and did find strong positive trends in SOS and POS in areas prone to multiple annual cycles, but overall found that both varying numbers of cycles and small amplitude cycles had more limited extent than could easily explain the large changes in trend we found.

Inherently, the scale of near-surface PhenoCam observations does not match those of remote sensing pixels, which include heterogenous vegetation and even land cover class [36,38,76]. The mismatch can be substantial even for our comparisons of PhenoCam with MODIS-based 250 or 500 m spatial resolution products versus CMGLSP at ~5 km resolution, in which the synthesis of large pixels could include billions of plants. LSP observations effectively provide only a broad picture of landscape phenology, and the scale of imagery has important implications for users. For example, CMGLSP at ~5 km spatial resolution may not provide the spatial variability needed to assess the behavior of a white-tailed deer or other animals with home ranges within one or two pixels, whereas it may be useful for understanding movement patterns of long distance migrating animals such as eagles or

mule deer. Given the heterogeneity of vegetation within large pixels, we were surprised at the high agreement of CMGLSP with PhenoCam, at a spatial resolution of 0.05 degrees. The dynamics of LSP metrics in large pixels are complex and do not necessarily represent the average of smaller pixels [77]. For SOS, [78] found that the heterogeneity of landcover and SOS, as well as vegetation growth speed during spring, all influenced the scale effect. Variation in spatial footprints of PhenoCam data and resolution of LSP datasets could also add noise to the comparison of LSP and near-surface datasets. New fine-resolution datasets, such as daily 30 m EVI products developed using fusion between MODIS and Landsat [79,80] or combinations of Landsat and Sentinal-2 imagery to increase temporal resolution and enable 30 m phenology retrievals [81], could better match the scale of reference data and user analysis objectives.

Applied data users will benefit from future developments in processing capacity and dataset construction, a better understanding of phenology drivers, and greater understanding of how algorithms intersect with phenology predictions. The growing popularity of platforms able to process large data quickly (such as Google Earth Engine) may render daily, global datasets derived from the DLC method readily available in the future. In terms of improving LSP estimates, the use of mechanistic models to predict key metrics [82–84] may help to address data quality issues and discrepancies across land cover types and ecoregions. These models couple remote sensing data with local observations or other models of elevation, temperature, precipitation, and plant phenology to improve phenology and productivity metrics. While still limited, in the future these models may be developed into a single framework to derive more relevant phenology and productivity estimates across diverse regions using localized data. Similarly, an increased understanding of the drivers of phenological changes and variability on a local or regional scale may help users to better interpret and plan for long-term patterns of change. For example, [85] showed that temperature, growing-season precipitation, and snowpack had larger effects than most management and habitat treatments on annual phenology metrics in sagebrush communities in Southwestern Wyoming, but identified that treatments had some impact on IVI and late season phenology metrics. Finally, deeper examination of how processing and curve-fitting techniques influence the resulting NDVI/EVI time series and phenology metrics will improve understanding of how to interpret large differences in phenology that are predicted from LSP data.

## 5. Conclusions

Land surface phenology (LSP) data provide the means to assess large spatial and temporal patterns in environmental change and understand a variety of ecological questions, such as those related to surface energy balance and animal fitness, movement, and habitat use. The quality of these data depend on factors including cloud cover, atmospheric effects, processing algorithm, land cover type, spatial resolution and regionality. We found highly variable agreement between 10 leading LSP datasets and PhenoCam and outlined dataset choices based on spatial and temporal coverage, processing capability, and agreement across ecosystems, providing a basis for informed and justifiable decisions about data choices. Most datasets we examined and many of the highest-quality datasets are readily available and do not require users to extract metrics, as is often performed by individual users using raw NDVI or EVI values. Given the popularity and importance of phenology and productivity data in a wide range of research and management fields [61,86], a better understanding and justification for the use of certain datasets can only help to bolster the effectiveness and validity of results.

**Author Contributions:** Conceptualization, T.A.G., N.L.M., J.A.M., A.N.J. and G.W.C.; methodology, E.E.B., T.A.G. and N.L.M.; software, E.E.B. and N.L.M.; validation, E.E.B.; formal analysis, E.E.B. and N.L.M.; resources, E.E.B. and T.A.G.; writing—original draft preparation, E.E.B. and N.L.M.; writing—review and editing, T.A.G., N.L.M., J.A.M., A.N.J. and G.W.C.; visualization, E.E.B.; supervision, T.A.G.; project administration, T.A.G.; funding acquisition, T.A.G.; E.E.B., T.A.G., and N.L.M. contributed equally. All authors have read and agreed to the published version of the manuscript.

**Funding:** This work was supported by the U.S. Geological Survey Northern Rocky Mountain Science Center and the North Central Climate Adaptation Science Center. The Wyoming Landscape Conservation Initiative provides support to Chong and Johnston. The work of Ethan Berman was done under contract to the U.S. Geological Survey. Any use of trade, firm, or product names is for descriptive purposes only and does not imply endorsement by the U.S. Government.

**Acknowledgments:** We would like to thank the Wyoming Game and Fish Department, Bureau of Land Management Kemmerer, Montana Department of Fish, Wildlife and Parks, and the National Park Service for their support in the development of this research. In addition, we appreciate the help of Jill Randall, Troy Fieseler, Brent Jamison, Arvid Aase, Kelly Proffitt, Justin Gude, as well as David Wood and anonymous reviewers for their feedback and comments.

**Conflicts of Interest:** The authors declare no conflict of interest. The external funders had no role in the design of the study; in the collection, analyses, or interpretation of data; in the writing of the manuscript, or in the decision to publish the results.

## Appendix A

Detailed table of phenology and productivity metrics along with formulas for metric calculations where not readily available.

**Table A1.** Detailed table of phenology and productivity metrics.

| Dataset | AVHRRP | eMODIS | MCD12Q2 | CMGLSP | VIPPHEN-EVI2 | VIPPHEN-NDVI | NPN First Leaf Spring Index | Landsat NPP | MODIS NPP | DLC | PhenoCam |
|---|---|---|---|---|---|---|---|---|---|---|---|
| Input data | Calibrated radiance data (level 1-B). | Calibrated radiance data (level 1-B). Aqua only | Time series of NBAR-EVI2 | AVHRR and MODIS EVI2 | AVHRR and MODIS EVI2 | AVHRR and MODIS NDVI | Temperature, weather events, in situ observations of one lilac and two honeysuckle species | MODIS NDVI composites, meteorological inputs, biome specific contraints | MODIS NDVI composites, meteorological inputs, biome specific contraints | MOD09Q1 surface reflection 8 day composites | RGB photo time series |
| Raw data processing details | Cleaned, NDVI calculated, and combined into 7 day composites using max NDVI and quality | Cleaned, NDVI calculated, and combined into 7 day composites using max NDVI and quality | Eliminating outliers and filling dormant period values | 2 band EVI (EVI2) determination from raw data. Smoothed with Savitsky-Golay filter | Data are smoothed with 2 step filter and 3 year moving window approach, AVHRR and MODIS continuity is based on linear regressions between overlapping data. | Data are smoothed with 2 step filter and 3 year moving window approach, AVHRR and MODIS continuity is based on linear regressions between overlapping data. | Calculated at PRISM minimum and maximum temperature values reach a "stable" state. | Daily GPP and maintenance respiration values calculated | Daily GPP and maintenance respiration values calculated | Cleaned, NDVI calculated | GCC |
| Metric extraction details | Did not receive information | SOS NDVI value compared with average of previous 36 days and SOS is when a trend shift is identified. Opposite for EOS | Cubic spline fit, SOS is 15% of AVI, EOS is 15% of AVI green-down | phenological changes are identified using hybrid piecewise logistic model | Phenological changes are identified using half-max method (at only 35%) | Phenological changes are identified using half-max method (at only 35%) | Average of the first day each year that the properties of the three individual species models are met (SOS only) | NPP extracted from daily GPP, maintenance respiration, and annual growth respiration | NPP extracted from daily GPP, maintenance respiration, and annual growth respiration | 2 piece logistic model. SOS/EOS are local min/max from 2nd derivatives | PELT method, AIC-selected smoothing spline, SOS at 10% of AVI, EOS at 10% of green-down curve |
| Spatial resolution | 1 km | 250 m | 500 m | 0.05° | 0.05° | 0.05° | 0.0417° | 30 m | 250 m | 250 m | Point |
| Temporal coverage | 1989–2014 | 2001–2018[1] | 2001–2017 | 1982–2016 | 1981–2016 | 1981–2016 | 1981–2019 | 1986–2018 | 2001–2018 | 2000–present | 2000–present |
| Accessible online? | yes | yes | yes | no | yes | yes | yes | yes | yes | yes | yes |
| Significant post-processing time requirement? | no | no | no | no | no | no | no | no | no | yes | no |
| Under production? | no | yes | yes | yes | no | no | yes | yes | yes | yes | yes |
| Spatial coverage | CONUS | CONUS | Global | Global | Global | Global | CONUS | CONUS | CONUS | Global | U.S. and Canada [2] |
| Years acquired | 2002–2014 | 2002–2014 [1] | 2002–2014 | 1982–2016 | 1982–2016 | 1982–2016 | 1982–2016 | 2002–2014 | 2002–2014 | 2002–2014 | 2002–2014 |
| Reference | [43] | [44] | [45] | [46] | [47] | [47] | [41] | [42] | [42] | [23] | [48] |
| Retrieval location | https://www.usgs.gov/land-resources/eros/phenology | https://www.usgs.gov/land-resources/eros/phenology | https://earthdata.nasa.gov/ | xiaoyang.zhang@sdstate.edu | https://earthdata.nasa.gov/ | https://earthdata.nasa.gov/ | https://www.usanpn.org/data/spring_indices | https://www.ntsg.umt.edu/project/landsat/landsat-productivity.php | http://files.ntsg.umt.edu/data/NTSG_Products/MOD17/MODIS_250/ | https://earthdata.nasa.gov/ | phenocamr |
| Metrics available or calculated by authors? (A is available, C is calculated, N is not available/used) | | | | | | | | | | | |
| SOS | A | A | A | A | A | A | A | N | N | C | C |
| PIRGd | C | C | C | C | C | C | N | N | N | C | C |
| POS | A | A | A | A | A | A | N | N | N | C | C |
| EOS | A | A | A | A | A | A | N | N | N | C | C |
| LOSp | C | C | C | C | C | C | N | N | N | C | C |
| LGS | A | A | C | C | A | A | N | N | N | C | C |
| IVI | A | A | A | A | A | A | N | A | A | C | C |
| PVI | A | A | C | A | A | A | N | N | N | C | C |
| AVI | A | A | A | C | A | A | N | N | N | C | C |

[1] Collection 6 of eMODIS starts in 2003, but Collection 5 is available from 2001. This analysis used Collection 5 for 2002; [2] Some sites in Panama, Hawaii, and Europe.

Metrics reflecting different components of forage phenology and productivity were derived (or readily provided) from the 10 datasets used in this analysis. In total we assessed ten different metrics, six related to phenology and three related to productivity. Many of the datasets provide the desired metrics, and when available these values were used. In other cases, the metrics were derived using the following equations (with the exception of the Bischof dataset, for which methods can be found in [23]:

$$PIRGd = SOS + \frac{POS - SOS}{2} \, (assuming \; logistic \; growth) \tag{A1}$$

$$POS = MAX_{i=SOS}^{EOS} Greenness_i \tag{A2}$$

$$LOSp = POS - SOS \tag{A3}$$

$$LGS = EOS - SOS \tag{A4}$$

$$IVI = \sum_{i=SOS}^{EOS} Greenness_i \tag{A5}$$

$$PVI = AVI + SMV \tag{A6}$$

$$AVI = PVI - SMV \tag{A7}$$

where PIRGd is peak instantaneous rate of spring greenup date, SOS is start of spring date, POS is peak of season, MAX is the maximum value for the given period, EOS is end of season, greenness$_i$ is the daily value of greenness (whether EVI, NDVI, or GCC), LOSp is length of spring, LGS is length of growing season, IVI is integrated vegetation index, PVI is peak vegetation index, AVI is amplitude of vegetation index and SMV is spring minimum value. All dates were represented as Julian date for calculation Purp.

## Appendix B

R$^2$ and mean bias results of 10 datasets tested against PhenoCam observation. The number of observations is displayed in parenthesis.

**Table A2.** Overall results.

| R Squared (N) | Phenology Metrics | | | | | | Productivity Metrics | | |
|---|---|---|---|---|---|---|---|---|---|
| Dataset | SOS | PIRGd | POS | EOS | LOSp | LGS | IVI | PVI | AVI |
| AVHRRP | 0.07 (98) | 0.20 (98) | 0.33 (106) | 0.22 (79) | 0.02 (98) | 0.00 (72) | 0.00 (104) | 0.13 (106) | 0.29 (106) |
| CMGLSP | 0.37 (102) | 0.52 (101) | 0.39 (105) | 0.20 (104) | 0.03 (101) | 0.06 (100) | 0.03 (100) | 0.00 (105) | 0.00 (104) |
| Landsat_NPP | NA (NA) | NA (NA) | NA (NA) | NA (NA) | NA (NA) | NA (NA) | 0.10 (88) | NA (NA) | NA (NA) |
| MCD12Q2 | 0.35 (76) | 0.55 (76) | 0.54 (76) | 0.35 (76) | 0.02 (76) | 0.02 (76) | 0.01 (76) | 0.01 (76) | 0.02 (76) |
| MODIS_NPP | NA (NA) | NA (NA) | NA (NA) | NA (NA) | NA (NA) | NA (NA) | 0.00 (78) | NA (NA) | NA (NA) |
| NPN | 0.03 (104) | NA (NA) | NA (NA) | NA (NA) | NA (NA) | NA (NA) | NA (NA) | NA (NA) | NA (NA) |
| VIPPHEN_EVI2 | 0.26 (106) | 0.42 (106) | 0.32 (106) | 0.16 (106) | 0.01 (106) | 0.04 (106) | 0.15 (106) | 0.13 (106) | 0.34 (103) |
| VIPPHEN_NDVI | 0.36 (106) | 0.44 (106) | 0.37 (106) | 0.23 (106) | 0.09 (106) | 0.10 (106) | 0.16 (106) | 0.15 (106) | 0.26 (97) |
| DLC | 0.35 (106) | 0.50 (106) | 0.53 (106) | 0.45 (106) | 0.11 (106) | 0.09 (106) | 0.00 (106) | 0.05 (106) | 0.05 (106) |
| eMODIS | 0.26 (99) | 0.33 (99) | 0.25 (103) | 0.22 (101) | 0.05 (99) | 0.05 (97) | 0.02 (103) | 0.11 (106) | 0.19 (103) |
| **Mean Bias (N)** | **Phenology Metrics** | | | | | | | | |
| Dataset | SOS | PIRGd | POS | EOS | LOSp | LGS | | | |
| AVHRRP | 9.88 (98) | 15.92 (98) | 20.54 (106) | 37.92 (79) | 14.09 (98) | 43.44 (72) | | | |
| CMGLSP | 8.41 (102) | −0.38 (101) | −12.10 (105) | 0.52 (104) | −17.75 (101) | −3.78 (100) | | | |
| Landsat_NPP | NA (NA) | NA (NA) | NA (NA) | NA (NA) | NA (NA) | NA (NA) | | | |
| MCD12Q2 | −4.39 (76) | 0.83 (76) | 7.05 (76) | 19.92 (76) | 12.45 (76) | 24.32 (76) | | | |
| MODIS_NPP | NA (NA) | NA (NA) | NA (NA) | NA (NA) | NA (NA) | NA (NA) | | | |
| NPN | −19.10 (104) | NA (NA) | NA (NA) | NA (NA) | NA (NA) | NA (NA) | | | |
| VIPPHEN_EVI2 | −4.39 (106) | 2.69 (106) | 10.76 (106) | 23.58 (106) | 16.15 (106) | 27.97 (106) | | | |
| VIPPHEN_NDVI | −25.49 (106) | −4.16 (106) | 18.17 (106) | 32.98 (106) | 44.66 (106) | 58.47 (106) | | | |
| DLC | −5.63 (106) | −17.66 (106) | 21.32 (106) | 4.13 (106) | 27.95 (106) | 9.76 (106) | | | |
| eMODIS | 4.61 (99) | 15.74 (99) | 23.82 (103) | 23.26 (101) | 24.26 (99) | 25.15 (97) | | | |

**Table A3.** Grasslands (GR) results.

| R Squared (N) | Phenology Metrics | | | | | | Productivity Metrics | | |
|---|---|---|---|---|---|---|---|---|---|
| Dataset | SOS | PIRGd | POS | EOS | LOSp | LGS | IVI | PVI | AVI |
| AVHRRP | 0.10 (41) | 0.13 (41) | 0.32 (45) | 0.52 (38) | 0.03 (41) | 0.15 (34) | 0.30 (45) | 0.05 (45) | 0.32 (45) |
| CMGLSP | 0.38 (44) | 0.52 (44) | 0.55 (45) | 0.56 (44) | 0.00 (44) | 0.01 (43) | 0.00 (43) | 0.00 (45) | 0.05 (44) |
| Landsat_NPP | NA (NA) | NA (NA) | NA (NA) | NA (NA) | NA (NA) | NA (NA) | 0.15 (37) | NA (NA) | NA (NA) |
| MCD12Q2 | 0.45 (37) | 0.69 (37) | 0.78 (37) | 0.51 (37) | 0.00 (37) | 0.00 (37) | 0.00 (37) | 0.05 (37) | 0.40 (37) |
| MODIS_NPP | NA (NA) | NA (NA) | NA (NA) | NA (NA) | NA (NA) | NA (NA) | 0.00 (34) | NA (NA) | NA (NA) |
| NPN | 0.07 (45) | NA (NA) | NA (NA) | NA (NA) | NA (NA) | NA (NA) | NA (NA) | NA (NA) | NA (NA) |
| VIPPHEN_EVI2 | 0.41 (45) | 0.64 (45) | 0.69 (45) | 0.50 (45) | 0.04 (45) | 0.24 (45) | 0.03 (45) | 0.12 (45) | 0.48 (45) |
| VIPPHEN_NDVI | 0.35 (45) | 0.50 (45) | 0.53 (45) | 0.68 (45) | 0.00 (45) | 0.18 (45) | 0.00 (45) | 0.05 (45) | 0.43 (45) |
| DLC | 0.57 (45) | 0.69 (45) | 0.58 (45) | 0.72 (45) | 0.07 (45) | 0.28 (45) | 0.43 (45) | 0.04 (45) | 0.02 (45) |
| eMODIS | 0.48 (42) | 0.60 (42) | 0.56 (44) | 0.60 (44) | 0.08 (42) | 0.18 (42) | 0.07 (44) | 0.14 (45) | 0.13 (44) |

| Mean Bias (N) | Phenology Metrics | | | | | |
|---|---|---|---|---|---|---|
| Dataset | SOS | PIRGd | POS | EOS | LOSp | LGS |
| AVHRRP | −7.17 (41) | 4.32 (41) | 15.56 (45) | 62.47 (38) | 24.98 (41) | 74.35 (34) |
| CMGLSP | 9.68 (44) | 3.02 (44) | −3.22 (45) | 18.93 (44) | −11.32 (44) | 11.40 (43) |
| Landsat_NPP | NA (NA) | NA (NA) | NA (NA) | NA (NA) | NA (NA) | NA (NA) |
| MCD12Q2 | −8.57 (37) | −1.11 (37) | 7.35 (37) | 40.32 (37) | 16.92 (37) | 48.89 (37) |
| MODIS_NPP | NA (NA) | NA (NA) | NA (NA) | NA (NA) | NA (NA) | NA (NA) |
| NPN | −20.18 (45) | NA (NA) | NA (NA) | NA (NA) | NA (NA) | NA (NA) |
| VIPPHEN_EVI2 | −15.07 (45) | 0.31 (45) | 16.69 (45) | 46.44 (45) | 32.76 (45) | 61.51 (45) |
| VIPPHEN_NDVI | −33.00 (45) | −8.67 (45) | 16.67 (45) | 54.44 (45) | 50.67 (45) | 87.44 (45) |
| DLC | −16.04 (45) | −24.19 (45) | 19.71 (45) | 25.04 (45) | 36.76 (45) | 41.09 (45) |
| eMODIS | −5.43 (42) | 8.80 (42) | 20.64 (44) | 39.02 (44) | 30.45 (42) | 49.74 (42) |

**Table A4.** Shrublands (SH) results.

| R Squared (N) | Phenology Metrics | | | | | | Productivity Metrics | | |
|---|---|---|---|---|---|---|---|---|---|
| Dataset | SOS | PIRGd | POS | EOS | LOSp | LGS | IVI | PVI | AVI |
| AVHRRP | 0.01 (7) | 0.07 (7) | 0.00 (7) | 0.20 (4) | 0.03 (7) | 0.40 (4) | 0.02 (7) | 0.16 (7) | 0.08 (7) |
| CMGLSP | 0.22 (7) | 0.28 (7) | 0.28 (7) | 0.62 (7) | 0.12 (7) | 0.20 (7) | 0.01 (7) | 0.19 (7) | 0.10 (7) |
| Landsat_NPP | NA (NA) | NA (NA) | NA (NA) | NA (NA) | NA (NA) | NA (NA) | 0.00 (7) | NA (NA) | NA (NA) |
| MCD12Q2 | 0.00 (1) | 0.00 (1) | 0.00 (1) | 0.00 (1) | 0.00 (1) | 0.00 (1) | 0.00 (1) | 0.00 (1) | 0.00 (1) |
| MODIS_NPP | NA (NA) | NA (NA) | NA (NA) | NA (NA) | NA (NA) | NA (NA) | 0.01 (7) | NA (NA) | NA (NA) |
| NPN | 0.13 (7) | NA (NA) | NA (NA) | NA (NA) | NA (NA) | NA (NA) | NA (NA) | NA (NA) | NA (NA) |
| VIPPHEN_EVI2 | 0.35 (7) | 0.28 (7) | 0.50 (7) | 0.39 (7) | 0.59 (7) | 0.07 (7) | 0.09 (7) | 0.46 (7) | 0.14 (7) |
| VIPPHEN_NDVI | 0.31 (7) | 0.22 (7) | 0.00 (7) | 0.15 (7) | 0.17 (7) | 0.35 (7) | 0.00 (7) | 0.19 (7) | 0.12 (7) |
| DLC | 0.55 (7) | 0.53 (7) | 0.43 (7) | 0.54 (7) | 0.44 (7) | 0.26 (7) | 0.00 (7) | 0.14 (7) | 0.07 (7) |
| eMODIS | 0.68 (7) | 0.89 (7) | 0.85 (7) | 0.03 (6) | 0.52 (7) | 0.22 (6) | 0.05 (7) | 0.22 (7) | 0.11 (7) |

| Mean Bias (N) | Phenology Metrics | | | | | |
|---|---|---|---|---|---|---|
| Dataset | SOS | PIRGd | POS | EOS | LOSp | LGS |
| AVHRRP | 38.43 (7) | 30.64 (7) | 23.86 (7) | 24.50 (4) | −13.57 (7) | −34.00 (4) |
| CMGLSP | 25.43 (7) | 6.79 (7) | −10.86 (7) | 8.71 (7) | −35.29 (7) | −16.71 (7) |
| Landsat_NPP | NA (NA) | NA (NA) | NA (NA) | NA (NA) | NA (NA) | NA (NA) |
| MCD12Q2 | −11.00 (1) | −11.00 (1) | −10.00 (1) | −2.00 (1) | 2.00 (1) | 9.00 (1) |
| MODIS_NPP | NA (NA) | NA (NA) | NA (NA) | NA (NA) | NA (NA) | NA (NA) |
| NPN | −108.57 (7) | NA (NA) | NA (NA) | NA (NA) | NA (NA) | NA (NA) |
| VIPPHEN_EVI2 | 25.71 (7) | 20.14 (7) | 15.57 (7) | 7.86 (7) | −9.14 (7) | −17.86 (7) |
| VIPPHEN_NDVI | 22.00 (7) | 11.79 (7) | 2.57 (7) | −13.71 (7) | −18.43 (7) | −35.71 (7) |
| DLC | 7.29 (7) | −10.79 (7) | 6.14 (7) | −24.00 (7) | −0.14 (7) | −31.29 (7) |
| eMODIS | 23.00 (7) | 21.43 (7) | 20.86 (7) | 15.00 (6) | −1.14 (7) | −6.67 (6) |

**Table A5.** Deciduous/Broadleaf (DB) results.

| R Squared (N) | Phenology Metrics | | | | | | Productivity Metrics | | |
|---|---|---|---|---|---|---|---|---|---|
| **Dataset** | **SOS** | **PIRGd** | **POS** | **EOS** | **LOSp** | **LGS** | **IVI** | **PVI** | **AVI** |
| AVHRRP | 0.37 (23) | 0.50 (23) | 0.21 (27) | 0.10 (26) | 0.02 (23) | 0.05 (23) | 0.26 (25) | 0.10 (27) | 0.34 (27) |
| CMGLSP | 0.65 (24) | 0.37 (23) | 0.00 (26) | 0.01 (26) | 0.00 (23) | 0.00 (23) | 0.00 (23) | 0.23 (26) | 0.26 (26) |
| Landsat_NPP | NA (NA) | NA (NA) | NA (NA) | NA (NA) | NA (NA) | NA (NA) | 0.24 (20) | NA (NA) | NA (NA) |
| MCD12Q2 | 0.28 (26) | 0.31 (26) | 0.19 (26) | 0.52 (26) | 0.07 (26) | 0.29 (26) | 0.00 (26) | 0.13 (26) | 0.25 (26) |
| MODIS_NPP | NA (NA) | NA (NA) | NA (NA) | NA (NA) | NA (NA) | NA (NA) | 0.90 (10) | NA (NA) | NA (NA) |
| NPN | 0.90 (27) | NA (NA) | NA (NA) | NA (NA) | NA (NA) | NA (NA) | NA (NA) | NA (NA) | NA (NA) |
| VIPPHEN_EVI2 | 0.04 (27) | 0.45 (27) | 0.13 (27) | 0.21 (27) | 0.17 (27) | 0.01 (27) | 0.54 (27) | 0.00 (27) | 0.33 (27) |
| VIPPHEN_NDVI | 0.59 (27) | 0.33 (27) | 0.07 (27) | 0.21 (27) | 0.07 (27) | 0.13 (27) | 0.59 (27) | 0.16 (27) | 0.25 (23) |
| DLC | 0.28 (27) | 0.51 (27) | 0.77 (27) | 0.33 (27) | 0.45 (27) | 0.07 (27) | 0.22 (27) | 0.20 (27) | 0.32 (27) |
| eMODIS | 0.02 (25) | 0.36 (25) | 0.26 (26) | 0.59 (26) | 0.02 (25) | 0.03 (25) | 0.24 (26) | 0.05 (27) | 0.59 (26) |
| **Mean Bias (N)** | **Phenology Metrics** | | | | | | | | |
| **Dataset** | **SOS** | **PIRGd** | **POS** | **EOS** | **LOSp** | **LGS** | | | |
| AVHRRP | −13.61 (23) | −5.28 (23) | 0.30 (27) | 17.15 (26) | 18.65 (23) | 39.83 (23) | | | |
| CMGLSP | −19.79 (24) | −18.33 (23) | −28.92 (26) | −37.04 (26) | −2.13 (23) | −6.30 (23) | | | |
| Landsat_NPP | NA (NA) | NA (NA) | NA (NA) | NA (NA) | NA (NA) | NA (NA) | | | |
| MCD12Q2 | −15.58 (26) | −7.46 (26) | 1.65 (26) | 4.08 (26) | 18.23 (26) | 19.65 (26) | | | |
| MODIS_NPP | NA (NA) | NA (NA) | NA (NA) | NA (NA) | NA (NA) | NA (NA) | | | |
| NPN | −18.74 (27) | NA (NA) | NA (NA) | NA (NA) | NA (NA) | NA (NA) | | | |
| VIPPHEN_EVI2 | −17.56 (27) | −8.39 (27) | 1.78 (27) | −1.15 (27) | 20.33 (27) | 16.41 (27) | | | |
| VIPPHEN_NDVI | −49.63 (27) | −27.22 (27) | −3.81 (27) | 10.63 (27) | 46.81 (27) | 60.26 (27) | | | |
| DLC | −20.19 (27) | −27.54 (27) | 18.52 (27) | −11.85 (27) | 39.70 (27) | 8.33 (27) | | | |
| eMODIS | −8.08 (25) | −0.08 (25) | 8.92 (26) | 11.15 (26) | 18.00 (25) | 21.48 (25) | | | |

**Table A6.** Evergreen (EN) results.

| R Squared (N) | Phenology Metrics | | | | | | Productivity Metrics | | |
|---|---|---|---|---|---|---|---|---|---|
| **Dataset** | **SOS** | **PIRGd** | **POS** | **EOS** | **LOSp** | **LGS** | **IVI** | **PVI** | **AVI** |
| AVHRRP | 0.03 (25) | 0.03 (25) | 0.04 (25) | 0.45 (9) | 0.02 (25) | 0.49 (9) | 0.01 (25) | 0.01 (25) | 0.04 (25) |
| CMGLSP | 0.33 (25) | 0.40 (25) | 0.28 (25) | 0.11 (25) | 0.11 (25) | 0.08 (25) | 0.01 (25) | 0.13 (25) | 0.01 (25) |
| Landsat_NPP | NA (NA) | NA (NA) | NA (NA) | NA (NA) | NA (NA) | NA (NA) | 0.04 (22) | NA (NA) | NA (NA) |
| MCD12Q2 | 0.44 (11) | 0.54 (11) | 0.64 (11) | 0.31 (11) | 0.08 (11) | 0.07 (11) | 0.15 (11) | 0.91 (11) | 0.75 (11) |
| MODIS_NPP | NA (NA) | NA (NA) | NA (NA) | NA (NA) | NA (NA) | NA (NA) | 0.02 (25) | NA (NA) | NA (NA) |
| NPN | 0.00 (25) | NA (NA) | NA (NA) | NA (NA) | NA (NA) | NA (NA) | NA (NA) | NA (NA) | NA (NA) |
| VIPPHEN_EVI2 | 0.16 (25) | 0.06 (25) | 0.01 (25) | 0.00 (25) | 0.04 (25) | 0.06 (25) | 0.06 (25) | 0.01 (25) | 0.09 (22) |
| VIPPHEN_NDVI | 0.24 (25) | 0.19 (25) | 0.04 (25) | 0.17 (25) | 0.12 (25) | 0.09 (25) | 0.06 (25) | 0.04 (25) | 0.52 (20) |
| DLC | 0.10 (25) | 0.07 (25) | 0.00 (25) | 0.09 (25) | 0.00 (25) | 0.02 (25) | 0.04 (25) | 0.00 (25) | 0.16 (25) |
| eMODIS | 0.09 (23) | 0.04 (23) | 0.00 (24) | 0.09 (23) | 0.00 (23) | 0.00 (22) | 0.07 (24) | 0.00 (25) | 0.11 (24) |
| **Mean Bias (N)** | **Phenology Metrics** | | | | | | | | |
| **Dataset** | **SOS** | **PIRGd** | **POS** | **EOS** | **LOSp** | **LGS** | | | |
| AVHRRP | 55.56 (25) | 52.88 (25) | 51.20 (25) | 14.33 (9) | −3.36 (25) | −23.56 (9) | | | |
| CMGLSP | 29.44 (25) | 9.52 (25) | −9.40 (25) | 5.04 (25) | −37.84 (25) | −24.40 (25) | | | |
| Landsat_NPP | NA (NA) | NA (NA) | NA (NA) | NA (NA) | NA (NA) | NA (NA) | | | |
| MCD12Q2 | 32.64 (11) | 22.68 (11) | 13.73 (11) | −11.36 (11) | −17.91 (11) | −44.00 (11) | | | |
| MODIS_NPP | NA (NA) | NA (NA) | NA (NA) | NA (NA) | NA (NA) | NA (NA) | | | |
| NPN | 7.52 (25) | NA (NA) | NA (NA) | NA (NA) | NA (NA) | NA (NA) | | | |
| VIPPHEN_EVI2 | 24.52 (25) | 15.70 (25) | 7.88 (25) | 18.84 (25) | −15.64 (25) | −5.68 (25) | | | |
| VIPPHEN_NDVI | 3.96 (25) | 26.14 (25) | 49.32 (25) | 39.36 (25) | 46.36 (25) | 35.40 (25) | | | |
| DLC | 25.64 (25) | 2.06 (25) | 28.68 (25) | −5.60 (25) | 4.04 (25) | −31.24 (25) | | | |
| eMODIS | 32.35 (23) | 44.93 (23) | 47.17 (24) | 10.17 (23) | 27.17 (23) | −8.32 (22) | | | |

## Appendix C

Additional figures relating to the long-term trend analysis of CMGLSP, VIP_EVI, and NPN.

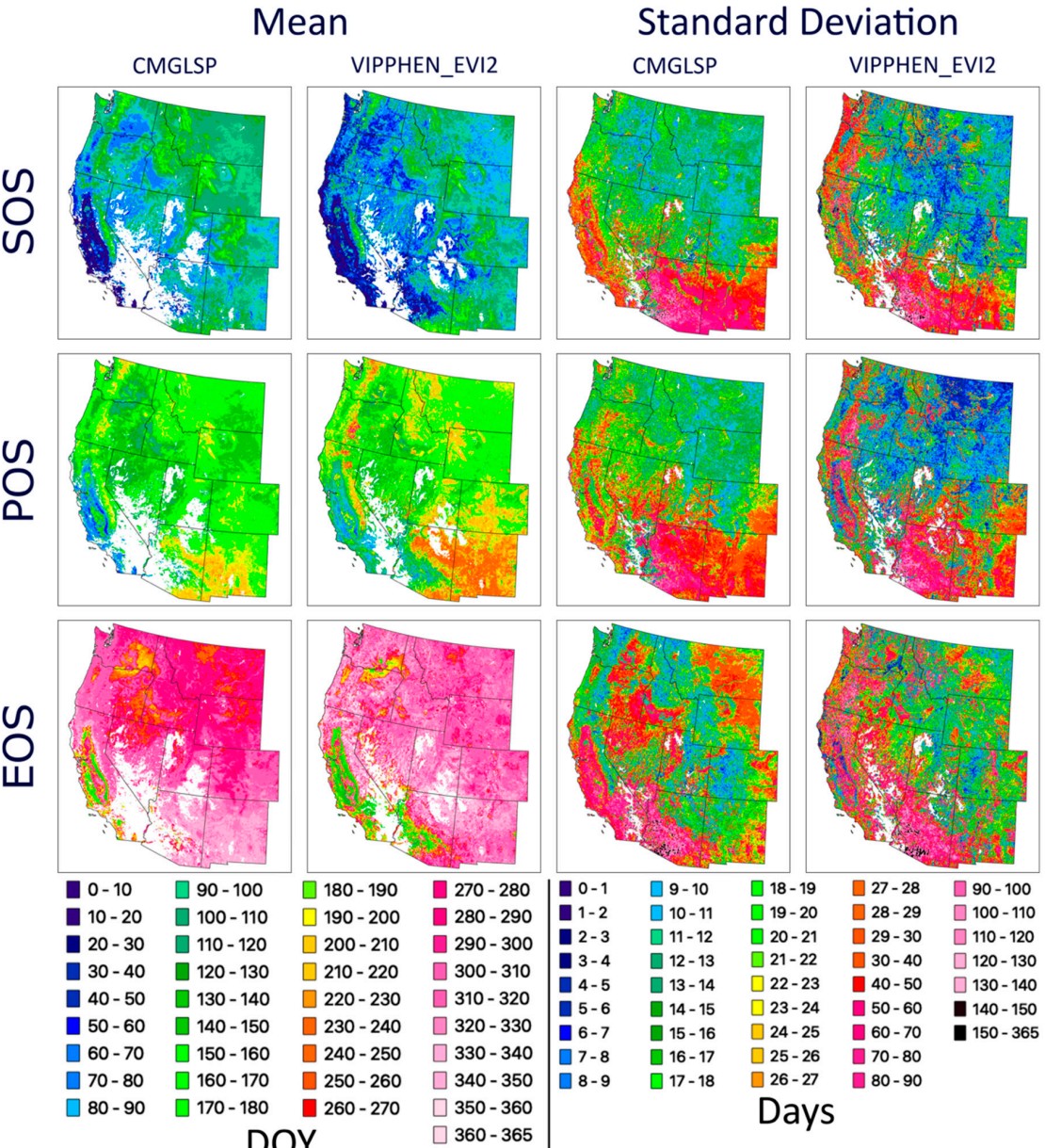

**Figure A1.** Mean and standard deviation of historical (1982–2016) data from CMGLSP and VIPPHEN_EVI2.

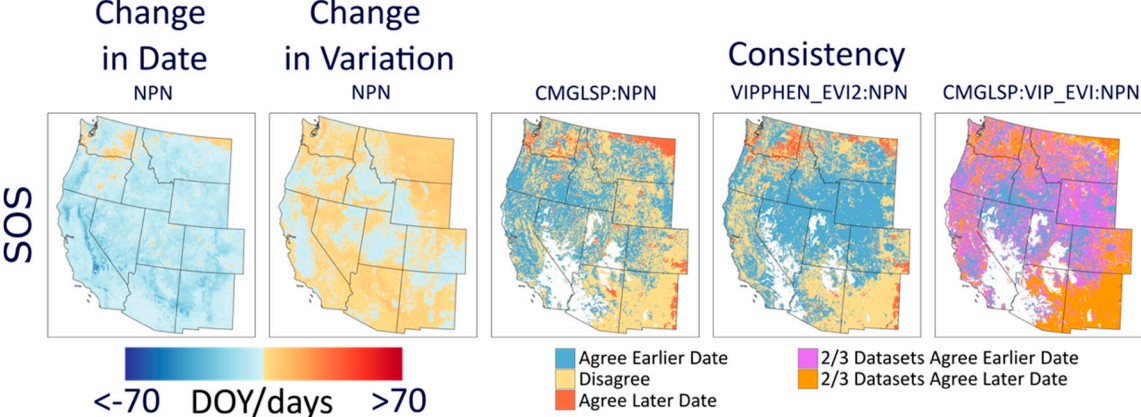

**Figure A2.** Change in SOS date and variation of NPN from 1982 to 2016 alongside consistency with CMGLSP and VIPPHEN_EVI2. Negative values correspond to earlier dates (or fewer days) and positive values to later dates (or more days). Consistency shows dataset agreement with regards to the direction of change. Change in date is the regressed trend slope multiplied by the number of years. Change in variation is the regressed trend slope of the absolute value of the residuals multiplied by the number of years.

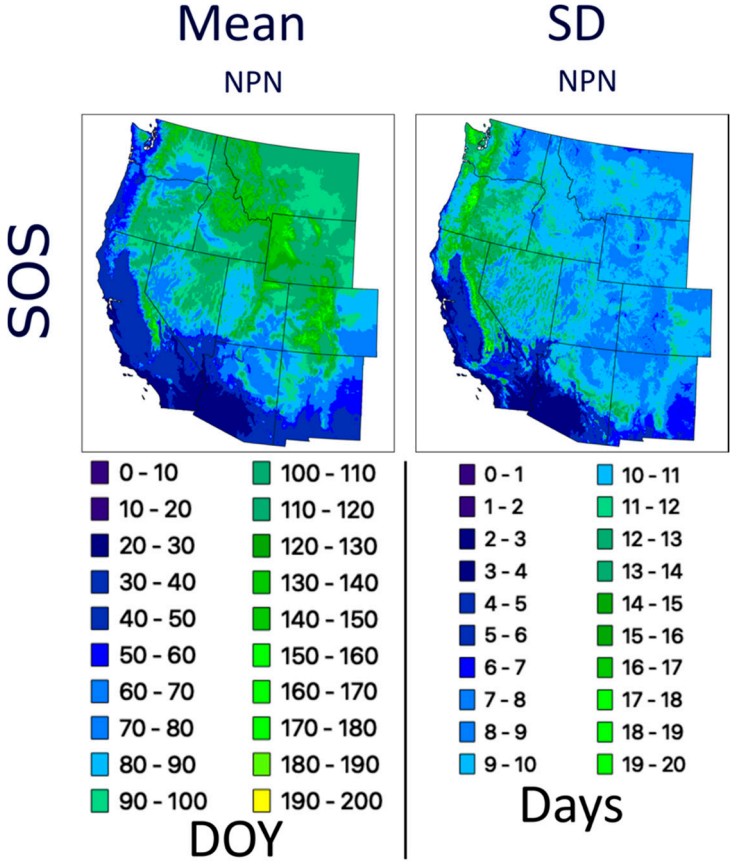

**Figure A3.** SOS Mean and standard deviation of historical (1982–2016) data from NPN.

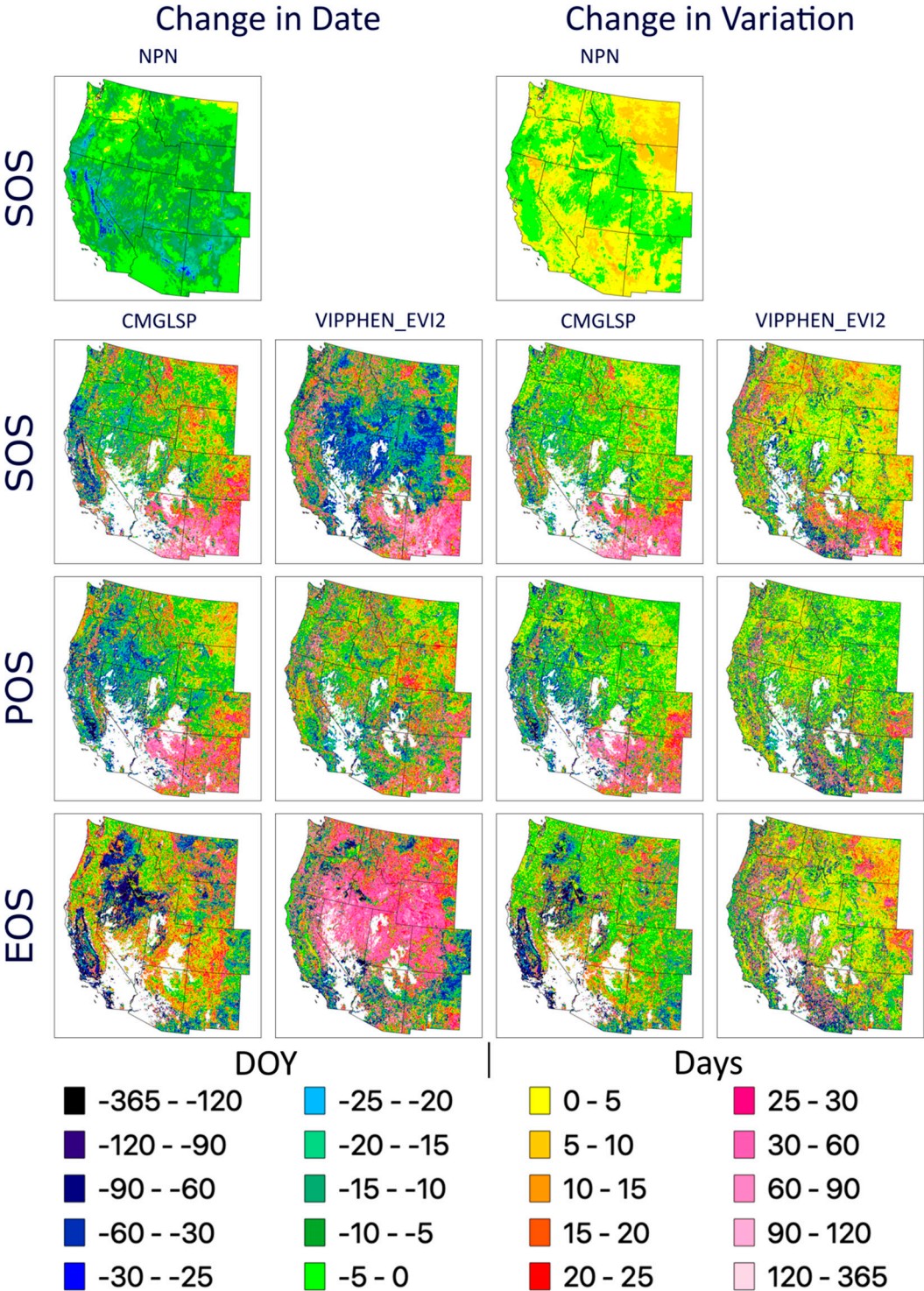

**Figure A4.** Change in date and variation of CMGLSP, VIPPHEN_EVI2, and NPN across three phenology metrics (only SOS for NPN) from 1982 to 2016. Shown here with a more robust color scheme to better indicate change. Negative values correspond to earlier dates (or fewer days) and positive values to later dates (or more days). Change in date is the regressed trend slope multiplied by the number of years. Change in variation is the regressed trend slope of the absolute value of the residuals multiplied by the number of years.

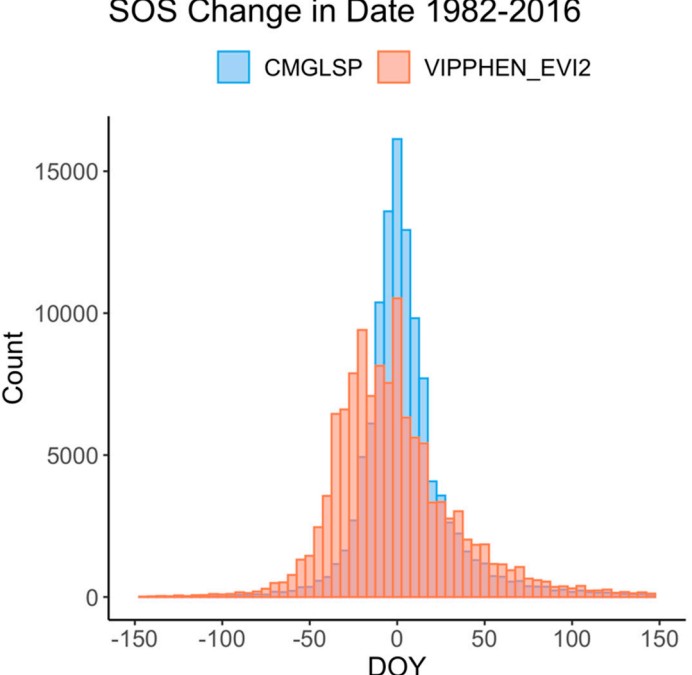

**Figure A5.** Histogram of CMGLSP and VIPPHEN_EVI2 SOS change in date from 1982 to 2016.

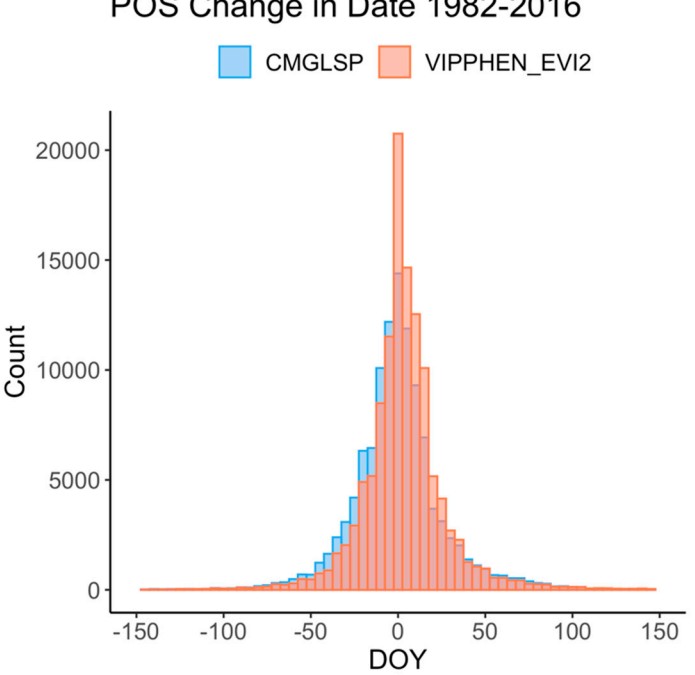

**Figure A6.** Histogram of CMGLSP and VIPPHEN_EVI2 POS change in date from 1982 to 2016.

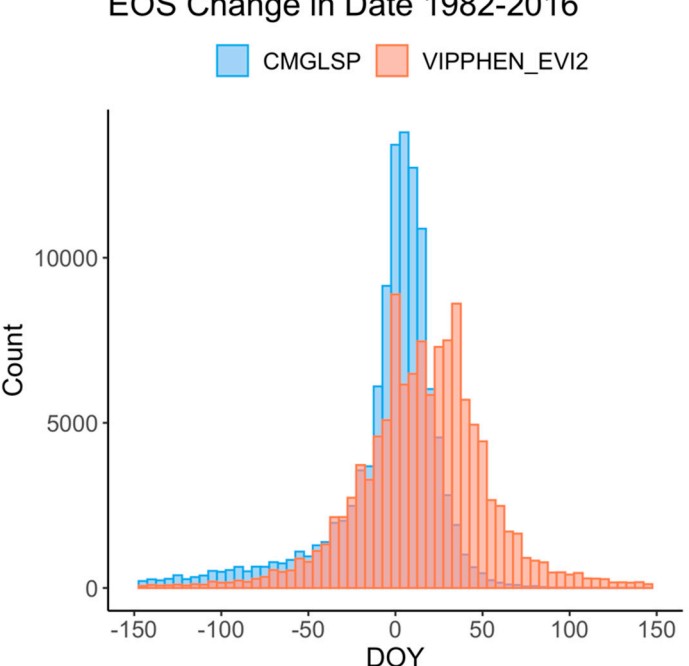

**Figure A7.** Histogram of CMGLSP and VIPPHEN_EVI2 EOS change in date from 1982 to 2016.

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
