# Peer review of "Comparative Quality and Trend of Remotely Sensed Phenology and Productivity Metrics across the Western United States"

_remotesensing, doi:10.3390/rs12162538_

Round 1
Reviewer 1 Report
This manuscript is well-executed and well-written, and will be helpful for scientists and managers for selecting the most appropriate phenological data for their purposes.
Author Response
Thank you very much for your time and feedback.
Reviewer 2 Report
Comparative quality and trend of remotely sensed forage phenology and productivity metrics across the western United States.
In this manuscript the authors assess a suite of remote sensing derived datasets, often used in land surface phenology (LSP) studies. Using a number of LSP metrics the authors compare these datasets to in situ PhenoCam measurements. The goal of the assessment is to provide users interested in LSP applications with detailed information about the range of available datasets enabling informed decision making regarding dataset selection. This is a valuable endeavor, given the ecological importance of LSP studies and the numerous applications across a range of fields. In its current form however, this manuscript is not suitable for publication and major revisions are recommended. In particular, the statistical analysis (merely using R2 and mean bias) to assess these data is not justified, and is fundamentally flawed, given the types of data and expected dynamics being compared. Without substantial reworking of this analysis, providing a more robust analysis, publishing this manuscript in Remote Sensing is not recommended. Below are a series general comments and recommendations, addressing issues with the overall manuscript along with a series of specific comments and questions.
General Comments:
- The writing (use of english, proper grammar etc.) is generally fine. However, the manuscript is still challenging to read, particularly the Materials and Methods, and Results sections. There are many acronyms and a lot of comparisons. It is too easy for a reader to get lost. Care should be taken, to lay out the materials, methods and results in a more systematic way. Hopefully, some of the specific comments below will clarify this point.
- Forage: The authors use ‘forage’ quality/quantity as the key ecosystem process related to LSP. It is mentioned specifically in the title. However, forage quality/quantity is one of many components related to phenology. If this study was specifically examining the phenology of rangelands (shrubs and grasses) this focus on forage could be justified. However, as the goal of the paper is a general assessment of LSP datasets and metrics across land cover classes and ecosystems, forage is better suited to be used as an example application rather than the focus.
- In Lines 85 -97, the authors provide a brief explanation of why differences may exist between LSP datasets and ground, or near-surface observations. The explanation provided however only seems to focus on the issue of pixel resolution (although it is not clear that this is what is being described). Given the purpose of this paper, a much deeper analysis of why differences may exist is warranted, of which pixel resolution is an important component. From a remote sensing perspective, other components include but are not limited to, overpass interval of the satellites, spectral resolution of the remote sensing data, cloud and atmospheric correction utilized and accuracy assessments, gap filling and smoothing methodologies of the products. In a similar vein, a description of the potential mismatch between PhenoCam observations and remote sensing datasets should be included. For example, how well does a PhenoCam FOV actually match the pixel being used, based on camera location?
- The statistical analysis utilized is very weak. No justification is provided for using R2 and mean bias as the two metrics by which to assess these datasets. In fact, I am not convinced that these two metrics are the appropriate metrics by which to analyze these data to answer the questions the authors are posing. The analysis is comparing estimates for SOS, PIRGd, POS, EOS, LOSp and LGS derived from PhenoCams and from Remote Sensing products. The remote sensing products are generally derived from compositing methods that choose the best available pixel within a given time frame. In many instances, particularly with SOS metrics this corresponds to time periods (spring) with high cloud cover and other atmospheric contamination. Thus, I would expect the date from remote sensing products, associated with SOS or other LSP metrics to be somewhat randomly associated with the date from the Phenocam network. Even if it is highly accurate (e.g. within 1 week), it is no surprise that the scatter around that date is not necessarily linearly related. There are also no results presented showing the distributions of the actual data. (DoY phenocam vs DoY R.S. product for the various metrics).
Specific Comments:
- Title: Identifying forage here seems to limit the application of this manuscript.
- Line 48. Phenology and productivity are not the sole drivers of ecosystem function. Perhaps reword to make clear.
- Line 51. Consider revising the sentence beginning with, “For example, phonology…”. There is a lot to unpack in there and could be broken up to improve clarity.
- Line 57. Land surface phenology is defined as “the study of vegetation phenology and productivity from remote sensing,...”. First, I suggest using a different word to define phenology than phenology. Second, while productivity is related to phenology, LSP is not the study of productivity. Perhaps refer to the National Phenology Network site for a precise definition.
- Line 69. Sentence beginning with “However, few studies…” I am not sure as to the point of this observation in this manuscript as this manuscript does not address this.
- Line 89. Simply cite [32,35]. There is no need to add ‘(but see)’.
- Lines 90-93. It is unclear to me what is being described by ‘LSP metrics synthesize information from millions…” Are the authors describing an inherent challenge with remote sensing, in that pixel level observations are the aggregation of all the reflectances within a given pixel?
- Line 93. Differences in the processing algorithms are mentioned as a possible reason for introduced bias. The authors should at least give some key examples of components in the processing algorithms that are potential culprits.
- Lines 116 - 123: The description of optical satellite imagery, vegetation indices and how it relates to extracting phenology metrics could be substantially improved.
- Figure 2: The difference between (8) PVI and (9) AVI is subtle, especially in this figure. The placement of 8 does not make it clear that PVI is any different from AVI. It may be worth clarifying the difference in these two metrics (mathematically and in their application towards understanding productivity).
- Lines 129-131. Authors should identify key characteristics of the MODIS and AVHRR satellites/sensors that could lead to differences in the derived datasets.
- Lines 129 - 146. It is not ideal to refer to Appendix A for key methodological information on the datasets. I also understand that describing them in full in the methods may not be efficient. Also, a better description of the MOD09Q1 processing is necessary, as it is somewhat more complex. Finally in Appendix A itself, it is unclear what processing was done by the authors and what processing is the standard processing of the products. Finally, many of these datasets include quality information which can be used to filter out poor quality data. It is unclear if this was done. Including data flagged as low quality could greatly reduce the R2 and mean bias.
- Line 152: Perhaps add some more details about the PhenoCam sites (e.g. how many sites within each land cover class). While it is in the results, it should be provided in methods.
- Line 173 -179. The authors should provide justification as to their sole use of R2 and mean bias to assess these datasets as opposed to other, potentially more robust statistical methods.
- Line 189. How is variation within a pixel calculated? Is this through time, all observations of the same pixel through time. If so, how is variability change through time assessed, some sort of moving window? These methods need to be clarified.
- Results. I leave it to the authors to address the questions to the statistical methods presented above. However, as stated, figures of the actual distributions of data would be helpful to understand the results and interpretations presented.
- Discussion: The general form of the discussion is good. Some points regarding selection of LSP dataset may need to be altered if statistical approach is altered.
Author Response
Please see the attachment. Thank you for your time and feedback.

Reviewer 3 Report
This study analyses 10 different satellite derived products (both greenness and productivity) and compare these datasets against Phenocam derived data. The authors additionally assess coherence between the two long-term products in the detection of phenological trends nearly in the last 40 years. The study sheds some light onto the pletora of available RS products and may serve as guide for phenologists.
The study is well designed, with sound and well explained methodology, and a thorough and effective discussion. For these reasons, it deserves publication in RS.
Minor points follow.
Fig3 and 4 captions: Recall to the reader the meaning of the phenology metrics acronyms for a self-explaining caption
Fig 5: Can the change in variation be interpreted as a change in susceptibility to interannual variability of meteorological drivers? Or is it due to an increase in extreme events? It would be nice to have a short paragraph in the discussion regarding this aspect.
Discrepancy between trends computed on different products and disturbance.
This is an important point that should be further expanded in the discussion: how important is to account for abrupt changes, which, by definition, is not a trend. Methods to detect abrupt disturbances (breakpoints, BFAST, etc.) must be discussed. An opinion by the author whether disturbed pixels should be included or excluded in a trend analysis would also be interesting.
L389-394: The multiple peak feature is an important point. Also here, an approach as BFAST or similar can search for abrupt changes also in the shape of the seasonal cycles, thereby identifying e.g. missing peaks due to water stress or higher-than-average number of peaks because of strong rain inputs.
Author Response

(The authors gave the same response as above.)

Round 2
Reviewer 2 Report
The authors addressed my concerns in the manuscript and provided adequate justification for my concerns which they did not modified.